# In-context Learning for Few-shot Multimodal Named Entity Recognition

**Chenran Cai**[1,2], **Qianlong Wang**[1,2], **Bin Liang**[1,2,4], **Bing Qin**[5],
**Min Yang**[6], **Kam-Fai Wong**[4], **Ruifeng Xu**[1,2,3] *

[1]Harbin Institute of Technology, Shenzhen, China
[2]Guangdong Provincial Key Laboratory of Novel Security Intelligence Technologies
[3]Peng Cheng Laboratory, Shenzhen, China
[4]The Chinese University of Hong Kong, Hong Kong, China
[5]Research Center for SCIR, Harbin Institute of Technology, Harbin, China
[6]SIAT, Chinese Academy of Science, Shenzhen, China
crcai@stu.hit.edu.cn, qlwang15@outlook.com, bin.liang@cuhk.edu.hk,
qinb@ir.hit.edu.cn, min.yang@siat.ac.cn, kfwong@se.cuhk.edu.hk,
xuruifeng@hit.edu.cn

## Abstract

Thanks in part to the availability of copious annotated resources for some entity categories, existing studies have achieved superior performance in multimodal named entity recognition (MNER). However, in the real-world scenario, it is infeasible to enumerate all entity categories in advance. Therefore, in this paper, we formulate a new few-shot multimodal named entity recognition (FewMNER) task, which aims to effectively locate and identify named entities for a text-image pair only using a small number of labeled examples. Further, we explore the merit of in-context learning (ICL) and propose a novel framework to deal with FewMNER, where three points are taken into account: *i.e.*, converting visual modality, selecting useful examples, and designing an effective task demonstration. Specifically, we first employ an image caption model to convert images into textual descriptions, enabling large language models to absorb information from visual modality. Then, we use the ranking of the sum of similarity rankings from both text and image modalities to select k-nearest examples, which form a demonstration context. Finally, we utilize the MNER definition and the meaning of each entity category as effective instruction. Extensive experimental results demonstrate that our framework outperforms baselines under several few-shot settings.

## 1 Introduction

Multimodal Named Entity Recognition (MNER) aims to identify named entities of different categories from the text with extra image assistance. Consider the example in Figure 1(a), we need to recognize three named entities from the text,

---
* Corresponding author

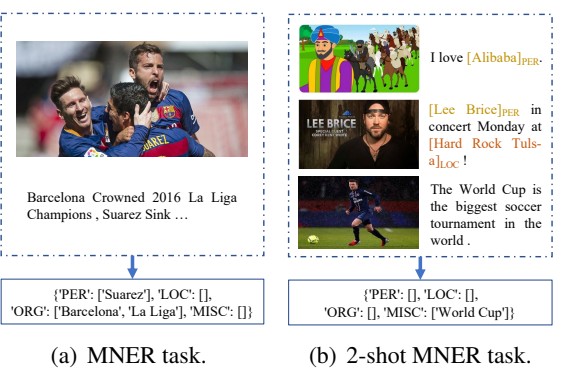

Figure 1: Illustration of MNER and 2-shot MNER tasks.

"*Suarez*" (PER), "*Barcelona*" and "*La Liga*" (ORG), to finish the MNER task. Most existing methods commonly employ pre-trained models followed by fine-tuning to accomplish the MNER task and achieve superior performance (Lu et al., 2018; Yu et al., 2020; Zhang et al., 2021; Chen et al., 2022b). In terms of existing research efforts, their superior performance generally relies on sufficient annotated data, which is time-consuming and labor-intensive. In addition, in practice, entity categories will continue to emerge rather than remain fixed. Therefore, it is impractical to define all entity categories in advance.

To address these issues, motivated by the few-shot Named Entity Recognition (FewNER) task that involves learning unseen entity categories from a small number of labeled examples (Fritzler et al., 2019), we extend the MNER task to the few-shot field, named the few-shot multimodal named entity recognition (FewMNER) task, which aims to locate and identify named entities for a text-image pair only using a small number of labeled examples. As illustrated in Figure 1(b), the 2-shot MNER task

aims to accomplish the MNER task based on two labeled text-image pair examples.

Further, to address the FewMNER task, we propose leveraging the powerful in-context learning (ICL) capability of the large language model (LLM). Specifically, we argue that this paradigm can provide a promising direction for solving the FewMNER task by learning from a few examples in context without training. However, there are three problems while solving FewMNER using the in-context learning paradigm: (*i*) For the FewMNER task, each sample is represented by textual and visual modalities, while the input of LLM is limited to natural language. Thus, we first require seeking ways to convert visual modality into natural language form. (*ii*) The key to performing ICL is to select a few examples to form a demonstration context. Although there are some example selection studies (Chen et al., 2022a; Min et al., 2022) targeting text classification tasks, selecting some useful examples for ICL in multimodal scenarios has not been approached. (*iii*) In addition, good demonstration designing precisely is essential to obtain satisfactory performance. Unlike simple classification tasks, the task instruction and output format of MNER need to be constructed according to the extractive nature.

To apply ICL to solve the FewMNER task, we propose corresponding solutions to the above-mentioned problems. First, we employ an image caption model (Wang et al., 2022a) to generate textual descriptions from images, which not only converts images into natural language form but also aligns image features into the text space. Second, for selecting examples, we design an efficient sorting algorithm based on image and text similarity ranks, which can mitigate the similarity bias caused by different modality models. Then, we utilize this algorithm to select top-$k$ examples with the highest similarity to the current test sample. Third, the demonstration design consists of two parts: instruction construction and demonstration construction (Dong et al., 2023). The former aims to inform LLM about the current task. To provide more detailed information, we add the description of entity category meaning to the instruction. The latter is to define the demonstration template and order selected examples into the demonstration template. For the demonstration template, it consists of three components: `image description`, `sentence`, and `output`, where `output` is the label information.

Then, we pack selected top-$k$ examples into the demonstration template in ascending order of similarity rank, such that the most similar example is nearest to the current test sample. Finally, we concatenate instruction, demonstration, and the test sample as the input and feed it into LLM to obtain the prediction `output`.

The contributions of this paper are as follows:

- We are the first to extend the MNER task to the few-shot field and explore the potential of the in-context learning paradigm for this task.

- To adapt the in-context learning paradigm to the FewMNER task, we address three related problems and propose a framework to accomplish this task.

- Through comparison with previous competitive methods, our framework exhibits a significant advantage in this task. We also conduct extensive analysis experiments to reveal the impact of various factors on its performance and provide novel insights for future research.

## 2 Related Work

### 2.1 Multimodal Named Entity Recognition

Multimodal Named Entity Recognition (MNER) aims to discover named entities in the unstructured text and classify them into pre-defined types with the help of an auxiliary image. Existing studies could be divided into two categories: *cross-modal interaction-based methods* and *image conversion-based methods*. The former tends to carry out cross-modal interaction using an attention mechanism and to combine textual representation with image representation for MNER. For example, some studies (Lu et al., 2018; Moon et al., 2018; Zhang et al., 2018) first applied LSTM and CNN to extract text and image features, respectively. Then attention is adopted to fuse two modal features to derive textual representation in order to complete entity labeling. In addition to modeling the interaction between text and images, a few studies (Chen et al., 2022b; Zhang et al., 2021) leveraged the semantic correlation between tokens and object regions to derive the final token representations for MNER. The latter (Chen et al., 2021; Wang et al., 2022b) first aims at converting images and extracting textualized information from them such as captions in order to align image features to the text space. Then this textualized information derived from an

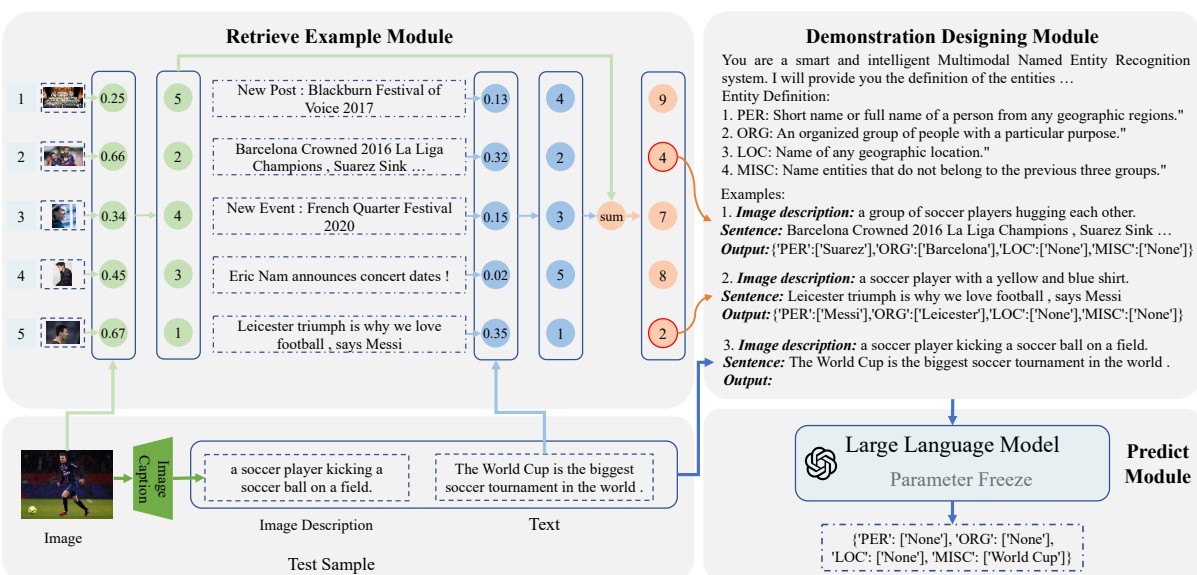

Figure 2: The architecture of our framework on the 2-shot MNER task.

image is concatenated with the input text to yield the final token representation for completion entity recognition. Despite their promising results, they generally depend on a large amount of annotated data, which is inadequate in generalizing the ability to locate and identify entities to unseen entity categories.

## 2.2 In-Context Learning

With the scaling of the pre-trained model from 110M parameters (Devlin et al., 2019) to over 500B parameters (Smith et al., 2022), the ability of the model has been greatly improved, especially the understanding ability, fluency, and quality of generation. Many studies have demonstrated that large language models (LLMs) have shown an in-context learning ability (Brown et al., 2020), which is learning from a few context examples without training. Although various LLMs (*e.g.*, GPT-3, ChatGPT) have been trained, they are all closed-source and only accessible internally or via paid API services. How to effectively utilize the in-context learning ability of LLMs is an important question. Recently, some studies (Sun et al., 2022; Hu et al., 2022; Zhang et al., 2022) treat LLMs as a service and utilize the in-context learning ability to finish the few-shot and even zero-shot tasks.

## 3 Methodology

### 3.1 Task Definition

Given a text $t$ and its correlated image $v$ as input, the fewMNER task only applies a small number of

labeled examples to detect a series of entities in $t$ and classify them into pre-defined categories.

Following most existing in-context learning work (Dong et al., 2023), we formulate this task as a generation task. A large language model $\mathcal{M}$ takes a generation sequence of the maximum score as prediction output conditioning on the context $C$. For the $k$-shot MNER task, $C$ contains instruction $I$ and $k$ examples, where $C = \{I, s(v_1, t_1, y_1), \ldots, s(v_k, t_k, y_k)\}$, $s$ is demonstration template and $\{y_1, \ldots, y_k\}$ is a set of free text phrases as the label. Therefore, for the given test sample $x = \{v, t\}$, the prediction output $\hat{y}$ can be expressed as:

$$\hat{y} = \max P_{\mathcal{M}}(y \mid C, x). \quad (1)$$

### 3.2 Overall Architecture

Figure 2 illustrates the overall architecture of our framework for the 2-shot MNER task, which contains three main components: (1) *Retrieve example module*, which utilizes $k$-nearest neighbors of text and image to select examples. (2) *Demonstration designing module*, which includes instruction construct and demonstration construct. (3) *Predict module*, which applies a large language model to generate prediction results without training.

### 3.3 Retrieve Example Module

Previous works (Rubin et al., 2022; Liu et al., 2022) have demonstrated that selecting similar examples to the current test sample can enhance the performance of LLM. However, these methods only con-

sider textual similarity scores, which are insufficient for the FewMNER task due to the multimodal nature of FewMNER. Besides, different modality models introduce bias (*i.e.*, different similarity score distributions for text and image (Peng et al., 2018)). To this end, we propose an efficient selection method based on text and image similarity ranks, which can mitigate the bias described above.

### 3.3.1 Image Similarity Rank

Given an test image $v_{test}$ and candidate set $\mathcal{D}$, where $\mathcal{D}$ contains $N$ text-image pair and $\mathcal{D} = \{(v_1, t_1), (v_2, t_2), ..., (v_N, t_N)\}$. We first adopt the pre-trained vision model ViT (Dosovitskiy et al., 2021) to obtain the representation of the whole image, including test image $v_{test}$ and candidate image set $\mathcal{D}_v = \{v_1, v_2, ..., v_N\}$:

$$\mathbf{H}_v = \text{ViT}(v_{test}), \quad (2)$$
$$\mathbf{V} = \text{ViT}(\mathcal{D}_v), \quad (3)$$

where $\mathbf{H}_v \in \mathbb{R}^{d_h}$ is the image representation of $v_{test}$ and $\mathbf{V} \in \mathbb{R}^{N \times d_h}$ is the embedding matrix of $\mathcal{D}_v$. Then, we calculate the cosine similarity of the test image representation $\mathbf{H}_v$ and the image representation of the whole candidate set $\mathbf{V}$, and record the rank of each candidate image set $\mathcal{D}_v$.

$$\mathbf{S}_v = \text{Cosine}(\mathbf{H}_v, \mathbf{V}), \quad (4)$$
$$\mathbf{R}_v = \text{Rank}(\mathbf{S}_v), \quad (5)$$

where $\mathbf{S}_v \in \mathbb{R}^N$, $\mathbf{R}_v \in \mathbb{R}^N$ and $\mathbf{R}_v^i \in [1, N]$.

### 3.3.2 Text Similarity Rank

Given a test text $t_{test}$, we first utilize the pre-trained language model such as MiniLM (Wang et al., 2020) as text extractor to map text $t_{test}$ and candidate text set $\mathcal{D}_t = \{t_1, t_2, ..., t_N\}$ into a $d_w$-dimensional embedding:

$$\mathbf{H}_t = \text{MiniLM}(t_{test}), \quad (6)$$
$$\mathbf{T} = \text{MiniLM}(\mathcal{D}_t), \quad (7)$$

where $\mathbf{H}_t \in \mathbb{R}^{d_w}$ is the sentence representation of $t_{test}$ and $\mathbf{T} \in \mathbb{R}^{N \times d_w}$ is the embedding matrix of $\mathcal{D}_t$. Then, we calculate the cosine similarity of the test text representation $\mathbf{H}_t$ and the text representation of the whole candidate set $\mathbf{T}$, and record the rank of each candidate text set $\mathcal{D}_t$.

$$\mathbf{S}_t = \text{Cosine}(\mathbf{H}_t, \mathbf{T}), \quad (8)$$
$$\mathbf{R}_t = \text{Rank}(\mathbf{S}_t), \quad (9)$$

where $\mathbf{S}_t \in \mathbb{R}^N$ and $\mathbf{R}_t \in \mathbb{R}^N$ and $\mathbf{R}_t^i \in [1, N]$.

### 3.3.3 Sorting Based on Both Similarity Ranks

According to the similarity rank results of image and text modalities $\mathbf{R}_v$ and $\mathbf{R}_t$, we sum two rankings and sort them to get the final ranking result.

$$\mathbf{R} = \text{Rank}(\mathbf{R}_v + \mathbf{R}_t), \quad (10)$$

where $\mathbf{R} \in \mathbb{R}^N$ and $\mathbf{R}^i \in [1, N]$. Compared with sorting based on the sum of image and text similarity scores, sorting based on both similarity ranks considers the bias introduced by different modality pre-trained models. Through analyzing the distribution of image similarity scores $\mathbf{S}_v$ and text similarity scores $\mathbf{S}_t$, we observe that the image similarity scores are generally higher than the text similarity scores. Sorting based on both similarity ranks can effectively address this issue. Finally, we take top-$k$ examples with the highest similarity ranking as selected examples.

$$\sigma = \text{Top-K}(\mathbf{R}), \quad (11)$$

where $\sigma$ are the indices of top-$k$ similarity ranking, and $\sigma = \{\sigma_1, ..., \sigma_k\}$.

### 3.4 Demonstration Designing Module

Following the in-context learning paradigm (Dong et al., 2023), it consists of two parts: instruction construction and demonstration construction.

### 3.4.1 Instruction Construction

We use the definition of the MNER task as the instruction, which helps LLM understand the current task and is shown as follows:

> *You are a smart and intelligent Multimodal Named Entity Recognition (MNER) system. I will provide you the definition of the entities you need to extract, the sentence from where your extract the entities, the image description from image associated with sentence and the output format with examples.*

To provide more detailed information for LLM, we describe the meaning of each entity category as follows:

> *1.PERSON: Short name or full name of a person from any geographic region; 2.ORGANIZATION: An organized group of people with a particular purpose, such as a business or a government department; 3.LOCATION: Names of any geographic location, like cities, countries, continents, districts, etc; 4.MISCELLANEOUS: Name entities that do not belong to the previous three groups PERSON, ORGANIZATION, and LOCATION.*

Finally, we concatenate the task and entity category definitions as instruction $\mathbf{I}$.

$$\mathbf{I} = \{task\ definition, category\ definition\}. \quad (12)$$

### 3.4.2 Demonstration Construction

As shown in the demonstration designing module in Figure 2, the demonstration template contains `image description`, `sentence`, and `output`. To obtain the `image description`, we employ the OFA model (Wang et al., 2022a) to convert images into text captions. The `sentence` is the original text input. The `output` is initially constructed by concatenating the entity and the category, taking the test sample in Figure 2 as an example, the initial `output` is "*World Cup is miscellaneous.*". However, this leads to disordered outputs, such as predicting categories that are not among the four predefined ones, despite the instruction specifying them. To address this issue, we adopt a dictionary-based format that explicitly defines the `output` structure as `{"PER": [], "ORG": [], "LOC": [], "MISC": []}`. We find that this approach effectively standardizes the output format[1].

Finally, the top-$k$ selected examples are fed into the demonstration template in ascending order such that the most similar example is nearest to the current test sample.

$$\mathbf{D} = \{s(\mathcal{D}_v^{\sigma_k}, \mathcal{D}_t^{\sigma_k}, \mathcal{D}_y^{\sigma_k}), ..., s(\mathcal{D}_v^{\sigma_1}, \mathcal{D}_t^{\sigma_1}, \mathcal{D}_y^{\sigma_1})\}. \tag{13}$$

### 3.5 Predict module

Given the instruction and demonstration, we concatenate them into the whole context $C$. Then, we feed context $C$ and test sample $\{v_{test}, t_{test}\}$ into LLM and select the most probable generated sequence as the predicted output.

$$C = \{\mathbf{I}, \mathbf{D}\}, \tag{14}$$
$$\hat{y} = \max P_{\mathcal{LLM}}(y \mid C, s(v_{test}, t_{test})). \tag{15}$$

Finally, we decode the prediction output $\hat{y}$ into a list according to the dictionary format and complete the $k$-shot MNER task.

## 4 Experiment

### 4.1 Dataset

We conduct experiments on two public multimodal named entity recognition (MNER) benchmark datasets, Twitter2015 and Twitter2017. Two MNER datasets are constructed by (Yu et al., 2020). Each example consists of a text and an associated image in the two MNER datasets. The statistics of two MNER datasets are shown in Table 1.

---

[1]To support this statement, we perform experiments comparing the two different output formats (detailed results in Appendix A.1).

| Entity Type | Twitter2015 | | | Twitter2017 | | |
|---|---|---|---|---|---|---|
| | Train | Dev | Test | Train | Dev | Test |
| Person | 2,217 | 552 | 1,816 | 2,943 | 626 | 621 |
| Location | 2,091 | 522 | 1,697 | 731 | 173 | 178 |
| Organization | 928 | 247 | 839 | 1,674 | 375 | 395 |
| Miscellaneous | 940 | 225 | 726 | 701 | 150 | 157 |
| Total | 6,167 | 1,546 | 5,078 | 6,049 | 1,324 | 1,351 |
| **Num of Tweets** | 4,000 | 1,000 | 3,257 | 3,373 | 723 | 723 |

Table 1: Statistics of two MNER datasets.

### 4.2 Experimental Settings

We randomly select 10, 50, 100, and all examples from the training set of two MNER datasets, respectively, and denote them as $\mathcal{D}_{10}$, $\mathcal{D}_{50}$, $\mathcal{D}_{100}$, and $\mathcal{D}_{all}$ sets. In this paper, we compare fine-tuning and few-shot methods with our framework. For fine-tuning methods, we utilize examples of the entire set to train models. For few-shot methods, we apply examples of the entire set as the train support and query sets and construct *4*-way $k$-shot setting for training. Here, the support and query sets of few-shot baselines used for training and inference have the same entity categories. For our framework, we select the $k$ examples from the entire set to perform the $k$-shot MNER task without training.

### 4.3 Model Settings

For the feature extraction model, we employ clip-vit-base-patch32[2] and all-MiniLM-L6-v2[3] to embed each image and text as a 512-dimensional and 768-dimensional embedding, respectively. For the image caption process, we employ the ofa-image-caption-coco-large-en[4] model to generate image description. For LLM, we choose the gpt-3.5-turbo (*i.e.*, ChatGPT) as the backbone of our framework.

### 4.4 Comparison Models

For fine-tuning methods, we adopt the following baselines: (1) UMT (Yu et al., 2020), which employs a transformer layer with a multimodal interaction module to capture the inter-modality dynamics between tokens and images for MNER; (2) UMGF (Zhang et al., 2021), which applies a unified multimodal graph approach to capture semantic relationships between tokens and visual objects and performs entity labeling; (3) HVPNeT (Chen et al., 2022b), which utilizes a hierarchical visual prefix fusion network for the visual-enhanced entity; (4)

---

[2]https://huggingface.co/openai/clip-vit-base-patch32
[3]https://huggingface.co/sentence-transformers/all-MiniLM-L6-v2
[4]https://modelscope.cn/models/damo/ofa_image-caption_coco_large_en

| Setting | Method | $\mathcal{D}_{10}$ | | | $\mathcal{D}_{50}$ | | | $\mathcal{D}_{100}$ | | | $\mathcal{D}_{all}$ | | |
|---|---|---|---|---|---|---|---|---|---|---|---|---|---|
| | | P(%) | R(%) | F1(%) | P(%) | R(%) | F1(%) | P(%) | R(%) | F1(%) | P(%) | R(%) | F1(%) |
| | | | | | | *Twitter2015* | | | | | | | |
| FT | UMT | 1.47 | 0.54 | 0.79 | 41.10 | 39.12 | 40.09 | 48.63 | 57.42 | 52.66 | 71.67 | 75.23 | 73.41 |
| | UMGF | 0.95 | 2.29 | 1.34 | 45.84 | 41.08 | 43.33 | 48.18 | 56.15 | 51.86 | 74.49 | 75.21 | 74.85 |
| | HVPNeT | 29.55 | 15.31 | 20.17 | 56.12 | 42.64 | 48.46 | 59.96 | 56.06 | 57.94 | 73.87 | 76.82 | 75.32 |
| | DebiasCL | 0.98 | 7.65 | 1.74 | 43.97 | 36.54 | 39.91 | 51.30 | 49.60 | 50.44 | 74.45 | 76.13 | 75.28 |
| 2-shot | ProtoBERT | - | - | - | 32.35 | 51.45 | 41.20 | 35.67 | 53.36 | 42.75 | - | - | - |
| | StructShot | - | - | - | 35.61 | 40.51 | 37.89 | 31.40 | 39.42 | 34.94 | - | - | - |
| | **Ours** | 45.03 | 63.72 | 52.76 | **46.39** | **63.82** | **53.73** | 45.96 | 64.95 | 53.83 | 47.73 | 65.25 | 55.13 |
| 4-shot | ProtoBERT | - | - | - | 32.40 | 53.04 | 40.23 | 35.63 | 56.34 | 43.64 | - | - | - |
| | StructShot | - | - | - | 30.26 | 37.20 | 33.34 | 26.66 | 35.13 | 30.14 | - | - | - |
| | **Ours** | 47.36 | 65.01 | 54.80 | **48.87** | **65.87** | **56.11** | 47.58 | 65.73 | 55.20 | 50.21 | 68.28 | 57.86 |
| 8-shot | ProtoBERT | - | - | - | 32.31 | 54.75 | 40.62 | 35.38 | 58.37 | 44.04 | - | - | - |
| | StructShot | - | - | - | 32.39 | 39.78 | 35.70 | 24.81 | 32.04 | 27.78 | - | - | - |
| | **Ours** | 50.44 | 64.39 | 56.57 | **49.94** | **64.13** | **56.15** | 48.72 | 64.35 | 55.45 | 51.24 | 67.20 | 58.14 |
| | | | | | | *Twitter2017* | | | | | | | |
| FT | UMT | 5.13 | 3.03 | 3.81 | 56.14 | 58.18 | 57.14 | 62.70 | 65.58 | 64.11 | 85.28 | 85.34 | 85.31 |
| | UMGF | 1.41 | 2.00 | 1.65 | 58.30 | 54.81 | 56.50 | 65.09 | 64.94 | 65.01 | 86.54 | 84.50 | 85.51 |
| | HVPNeT | 35.44 | 31.90 | 33.58 | 61.56 | 55.96 | 58.63 | 66.21 | 64.10 | 65.14 | 85.84 | 87.93 | 86.87 |
| | DebiasCL | 1.93 | 12.85 | 3.36 | 50.40 | 35.20 | 41.45 | 70.48 | 65.36 | 67.83 | 87.59 | 86.11 | 86.84 |
| 2-shot | ProtoBERT | - | - | - | 42.51 | 58.99 | 49.39 | 48.91 | 63.58 | 55.28 | - | - | - |
| | StructShot | - | - | - | 41.51 | 50.68 | 45.63 | 49.66 | 57.12 | 53.12 | - | - | - |
| | **Ours** | 61.30 | 72.09 | 66.26 | **62.40** | **71.72** | **66.74** | 64.79 | 73.43 | 68.84 | 65.82 | 74.54 | 69.91 |
| 4-shot | ProtoBERT | - | - | - | 42.50 | 61.56 | 50.27 | 49.32 | 66.37 | 56.58 | - | - | - |
| | StructShot | - | - | - | 40.40 | 50.47 | 44.72 | 47.26 | 56.26 | 51.35 | - | - | - |
| | **Ours** | 64.08 | 74.46 | 68.88 | **65.58** | **74.46** | **69.74** | 66.05 | 76.46 | 70.87 | 67.27 | 77.28 | 71.92 |
| 8-shot | ProtoBERT | - | - | - | 37.35 | 59.75 | 49.67 | 50.48 | 66.73 | 57.47 | - | - | - |
| | StructShot | - | - | - | 42.09 | 53.17 | 46.98 | 47.66 | 55.13 | 51.10 | - | - | - |
| | **Ours** | 65.33 | 73.35 | 69.11 | **66.76** | **73.58** | **70.00** | 69.24 | 77.13 | 72.97 | 69.00 | 78.09 | 73.26 |

Table 2: Main experiment results to compare fine-tuning and few-shot baselines with our framework in 2-shot, 4-shot, and 8-shot settings on $\mathcal{D}_{10}$, $\mathcal{D}_{50}$, $\mathcal{D}_{100}$, and $\mathcal{D}_{all}$ sets of Twitter2015 and Twitter2017 datasets. FT denotes the fine-tuning method. For $\mathcal{D}_{10}$ set, results with - due to *4-way 2/4/8-shot* setting more than number of $\mathcal{D}_{10}$ set. For $\mathcal{D}_{all}$ set, few-shot methods are similar to fine-tuning methods. Therefore, we use "-" to indicate that this is not a valid few-shot setting. To ensure the reliability of few-shot baselines, the results of ProtoBERT and StructShot are the average results of 100 runs.

DebiasCL (Zhang et al., 2023), which proposes a de-bias contrastive learning-based approach for MNER and studies modality alignment enhanced by cross-modal contrastive learning.

For the few-shot methods, we apply the following baselines: (1) ProtoBERT (Ding et al., 2021), which employs a prototypical network with a backbone of BERT encoder; (2) StructShot (Yang and Katiyar, 2020), which uses token-level nearest neighbor classification and structured inference.

### 4.5 Main Results

We report the main experimental results in Table 2 and draw the following conclusions.

(1) Our framework significantly outperforms the fine-tuning methods on $\mathcal{D}_{10}$, $\mathcal{D}_{50}$ and $\mathcal{D}_{100}$ sets (except for HVPNeT on $\mathcal{D}_{100}$ set of Twitter2015). For example, in terms of F1, our framework outperforms UMT by 55.78% and 65.30%, UMGF by 55.23% and 67.46%, HVPNeT by 36.40% and 35.53%, and DebiasCL by 54.83% and 65.75% on $\mathcal{D}_{10}$ set of two MNER datasets. These show that

our framework effectively exploits the in-context learning potential of large language models.

(2) Compared with few-shot baselines such as ProtoBERT and StructShot, our framework achieves the best results in all few-shot settings (*i.e.*, 2-shot, 4-shot, and 8-shot). This indicates that methods based on in-context learning are preferable in the FewMNER task, and thus exploring congenial methods based on in-context learning can lead to improved performance in this task.

(3) We observe that the performance of our framework improves as the size of $\mathcal{D}$ increases. This is because a larger retrieval set provides more opportunities for the test sample to find similar examples.

(4) Our framework still lags behind the fine-tuning methods under the $\mathcal{D}_{all}$ set.

### 4.6 Ablation Study

To analyze the impact of instruction and demonstration on the performance of our framework, we conduct ablation experiments and report detailed

| Methods | Twitter2015 | | | | | | Twitter2017 | | | | | |
|---|---|---|---|---|---|---|---|---|---|---|---|---|
| | 2-shot | | | 4-shot | | | 2-shot | | | 4-shot | | |
| | P(%) | R(%) | F1(%) | P(%) | R(%) | F1(%) | P(%) | R(%) | F1(%) | P(%) | R(%) | F1(%) |
| **Ours** | 46.39 | **63.82** | **53.73** | **48.87** | **65.87** | **56.11** | 62.40 | **71.72** | **66.74** | 65.58 | **74.46** | **69.74** |
| *w/o* **I** | 45.46 | 63.17 | 52.88 | 48.50 | 64.33 | 55.30 | 60.55 | 71.58 | 65.60 | 65.39 | 73.58 | 69.24 |
| *w/o* **D** | 43.35 | 40.01 | 41.61 | 43.35 | 40.01 | 41.61 | 63.44 | 53.29 | 57.92 | 63.44 | 53.29 | 57.92 |
| *w/ score* | **46.47** | 63.68 | **53.73** | 46.78 | 64.07 | 54.08 | **62.55** | 70.10 | 66.11 | 64.42 | 69.95 | 67.07 |

Table 3: Ablation study in 2-shot and 4-shot settings on $\mathcal{D}_{50}$ set. **I** and **D** denote instruction and demonstration, respectively. *w/ score* indicates that examples are selected based on the sum of image and text similarity scores.

| Modality | Twitter2015 | | | | | | | Twitter2017 | | | | | | |
|---|---|---|---|---|---|---|---|---|---|---|---|---|---|---|
| | Single Type (F1) | | | | Overall | | | Single Type (F1) | | | | Overall | | |
| | PER | LOC | ORG | MISC | P(%) | R(%) | F1(%) | PER | LOC | ORG | MISC | P(%) | R(%) | F1(%) |
| single | 77.06 | **67.08** | **39.86** | 17.48 | 47.45 | **66.85** | 55.50 | 88.96 | **72.34** | **67.27** | **18.18** | 64.29 | 74.39 | 68.98 |
| multi | **77.33** | 65.91 | 38.54 | **20.21** | **48.87** | 65.87 | **56.11** | **89.50** | 71.26 | 66.23 | 17.33 | **65.58** | **74.46** | **69.74** |

Table 4: Different modalities in 4-shot setting on $\mathcal{D}_{50}$ set.

| Methods | Twitter2015 | | | Twitter2017 | | |
|---|---|---|---|---|---|---|
| | P(%) | R(%) | F1(%) | P(%) | R(%) | F1(%) |
| random | 47.46 | 64.54 | 54.70 | 64.21 | 73.58 | 68.58 |
| dissimilar | 47.08 | 64.56 | 54.46 | 61.84 | 71.13 | 66.16 |
| similar | **48.87** | **65.87** | **56.11** | **65.58** | **74.46** | **69.74** |

Table 5: Different selecting example methods in 4-shot setting on $\mathcal{D}_{50}$ set.

| Methods | Twitter2015 | | | Twitter2017 | | |
|---|---|---|---|---|---|---|
| | P(%) | R(%) | F1(%) | P(%) | R(%) | F1(%) |
| descending | 48.61 | 65.29 | 55.73 | 65.14 | **74.98** | 69.71 |
| random | 48.43 | 65.42 | 55.66 | 65.49 | 74.32 | 69.62 |
| ascending | **48.87** | **65.87** | **56.11** | **65.58** | 74.46 | **69.74** |

Table 6: Different sorting example methods in 4-shot setting on $\mathcal{D}_{50}$ set.

results in Table 3. The results reveal that our framework achieves the best performance when combining instruction and demonstration, which suggests that both components are beneficial. Compared with removing instruction, removing demonstration leads to more performance degradation. This shows that the demonstration is crucial for our framework.

Furthermore, we also conduct ablation experiments on the way of selecting examples. Sorting based on the sum of image and text similarity ranks outperforms sorting based on the sum of image and text similarity scores (*i.e.*, *w/ score*). This is because the latter does not account for the bias introduced by different modality models.

## 4.7 Analysis

**Image Modality Analysis.** To explore the effect of image description on the FewMNER task, we conduct experiments with single-modality (*i.e.*, text) and multi-modality (*i.e.*, text and image) and show results in Table 4. For a fair comparison, both settings use the same instruction and demonstration, but the single-modality setting discards the `image description`. We observe that the multi-modality setting outperforms the single-modality setting, especially on the PER category. The reason is that the image caption model tends to generate sentences related to people, which provide useful cues for identifying PER category entities.

**Different Examples Analysis.** To analyze the impact of different examples, we compare three methods of examples selection (*i.e.*, similar, dissimilar, and random) in the 4-shot setting on $\mathcal{D}_{50}$ set. The similar method selects examples that have the highest similarity to the test sample, while the dissimilar method selects examples that have the lowest similarity. The random method selects examples uniformly at random. The results are shown in Table 5. We observe that the similar method achieves the best performance, followed by the random method, and the dissimilar method performs the worst. This indicates that selecting similar examples to form the demonstration is beneficial for the FewMNER task.

**Impact of Examples Sort.** To investigate the impact of example sort on performance, we utilize the same examples to compare three methods of sorting (*i.e.*, ascending, descending, and random) in the 4-shot setting on $\mathcal{D}_{50}$ set. The results are shown in Table 6. The ascending sort method, which places the most similar example nearest to the current test sample, outperforms the other methods. This suggests that ascending sort examples by their similarity can improve performance. The reason is that the most similar example can provide more relevant information for the current prediction.

**Impact of the Number of Examples.** To explore the impact of the number of examples on perfor-

| Setting | PER | | | LOC | | | ORG | | | MISC | | | Overall |
|---|---|---|---|---|---|---|---|---|---|---|---|---|---|
| | P(%) | R(%) | F1(%) | P(%) | R(%) | F1(%) | P(%) | R(%) | F1(%) | P(%) | R(%) | F1(%) | F1(%) |
| | | | | | | *Twitter2015* | | | | | | | |
| 2-shot | 70.84 | 79.45 | 74.90 | 66.37 | 64.99 | 65.67 | 24.96 | 61.03 | 35.43 | 14.42 | 24.86 | 18.26 | 53.73 |
| 4-shot | **72.67** | **82.63** | **77.33** | 66.67 | **65.16** | **65.91** | 27.30 | **65.55** | 38.54 | 16.65 | **25.69** | **20.21** | 56.11 |
| 8-shot | 72.14 | 80.71 | 76.18 | **66.99** | 64.70 | 65.82 | **27.79** | 63.53 | **38.67** | **16.68** | 21.69 | 18.86 | **56.15** |
| | | | | | | *Twitter2017* | | | | | | | |
| 2-shot | 90.75 | 83.74 | 87.10 | 69.66 | 69.66 | 69.66 | 56.87 | 74.43 | 64.47 | 10.88 | 19.75 | 14.03 | 66.74 |
| 4-shot | 92.01 | **87.12** | **89.50** | **72.94** | 69.66 | **71.26** | 57.84 | **77.47** | 66.23 | 14.17 | **22.29** | 17.33 | 69.74 |
| 8-shot | **92.44** | 84.70 | 88.40 | 70.17 | **71.35** | 70.75 | **59.53** | **77.47** | **67.33** | **15.56** | **22.29** | **18.32** | **70.00** |

Table 7: The detail metric (*i.e.*, P, R, and F1) for four categories in 2-shot, 4-shot, and 8-shot settings on $\mathcal{D}_{50}$ set.

| Setting | Twitter2015 | | | | | Twitter2017 | | | | |
|---|---|---|---|---|---|---|---|---|---|---|
| | $N_p$ | $B.(\%)\downarrow$ | $N_{cb}$ | $C.(\%)\downarrow$ | $F1(\%)\uparrow$ | $N_p$ | $B.(\%)\downarrow$ | $N_{cb}$ | $C.(\%)\downarrow$ | $F1(\%)\uparrow$ |
| 2-shot | 7038 | 45.15 | 3860 | 15.41 | 53.73 | 1553 | 28.59 | 1109 | 12.62 | 66.74 |
| 4-shot | 6896 | 42.92 | 3936 | **14.38** | 56.11 | 1534 | 25.55 | 1142 | **11.91** | 69.74 |
| 8-shot | 6570 | **41.35** | 3853 | 14.85 | **56.15** | 1489 | **23.50** | 1139 | 12.73 | **70.00** |

Table 8: Analysis of the wrong predictions in 2-shot, 4-shot, and 8-shot settings on $\mathcal{D}_{50}$ set. $N_p$, $B.$, $N_{cb}$, and $C.$ denote the number of predictions, proportion of boundary errors, correct number of boundaries, and proportion of category errors on the FewMNER task, respectively.

| Setting | Twitter2015 | | | Twitter2017 | | |
|---|---|---|---|---|---|---|
| | P(%) | R(%) | F1(%) | P(%) | R(%) | F1(%) |
| 0-shot | 43.35 | 40.01 | 41.61 | 63.44 | 53.29 | 57.92 |
| 2-shot | 46.39 | 63.82 | 53.73 | 62.40 | 71.72 | 66.74 |
| 4-shot | 48.87 | 65.87 | 56.11 | 65.58 | 74.46 | 68.74 |
| 8-shot | 49.94 | 64.13 | 56.15 | 66.76 | 73.58 | 70.00 |
| 16-shot | **50.83** | **64.86** | **56.99** | 68.04 | **74.69** | 71.21 |
| 32-shot | 49.35 | 64.50 | 55.92 | **69.31** | 74.38 | **71.76** |

Table 9: Different shot settings on $\mathcal{D}_{50}$ set.

mance, we conduct experiments with different the number of examples (*i.e.*, 0, 2, 4, 8, 16, 32-shot) on $\mathcal{D}_{50}$ set and show results in Table 9. On the Twitter2017 dataset, we observe that the F1 generally increases with the number of examples, and our framework achieves the best score with 32 examples. Comparing 0-shot with 32-shot, the latter outperforms the former by 13.84% in F1.

Recently, some works have attempted to explain the ICL capability of LLMs. Dai et al. (2023) interprets language models as meta-optimizers and views ICL as a form of implicit fine-tuning. This is consistent with our findings that more examples can enhance the performance, as more examples lead to more optimization steps. On the Twitter2015 dataset, we observe a similar trend as on the Twitter2017 dataset, but our framework achieves the best score with 16 examples. The reason is that increasing the number of examples may introduce more dissimilar examples. These fluctuations indicate that more examples can have a positive effect on performance if they are sufficiently similar.

**Error analysis.** In this section, we aim to analyze the factors that affect the performance of our framework. As shown in Table 7, we report the performance of four categories in 2-shot, 4-shot, and 8-shot settings on $\mathcal{D}_{50}$ set. We find that our framework performs poorly on MISC category, which is significantly lower than PER, LOC, and ORG categories. The reason is that MISC is a miscellaneous category, defined as name entities that do not belong to the previous three categories. The annotation of MISC category entities depends on the preference of annotators. Relying only on the in-context learning ability of LLM and a few examples is not sufficient to learn this preference.

Moreover, we analyze the boundary error and category error and perform a detailed analysis of wrong predictions. We classify wrong predictions into two types: boundary errors and category errors[5]. We count the number of errors for each category and report results in Table 8. We observe that increasing the number of examples significantly reduces boundary errors. Specifically, comparing 2-shot with 8-shot, the latter reduces the proportion of boundary errors by 3.80% and 5.09% on two datasets, respectively. Besides, increasing the number of examples does not reduce category errors. This is an interesting finding and demonstrates that more examples mainly improve the boundary ability of ICL, rather than category ability.

## 5 Conclusion

In this paper, we formulate multimodal named entity recognition as a few-shot learning problem,

---

[5]When the predicted entity is boundary-misspecified, we classify it as a boundary error. When the boundary of the entity is completely correct, but the category is incorrectly identified, we classify it as a category error.

named few-shot multimodal named entity recognition (FewMNER), to extend entity detection to unseen entity categories. To tackle FewMNER, we propose a framework based on in-context learning by addressing three problems. Experimental results show that our framework outperforms baselines in several few-shot settings. Moreover, we conduct analysis experiments and find that selecting similar examples, sorting them in ascending order, and using more examples improve the performance of in-context learning. We also perform error analysis and observe that increasing the number of examples reduces boundary errors but not category errors. These results provide novel insights for future work on the FewMNER task.

## Limitations

Although the proposed framework significantly outperforms several strong baselines on the FewMNER task, it suffers from the following limitations:

- In the case of using the full amount of data, our framework still lags behind fine-tuning methods. We can utilize some data to domain fine-tune LLM before applying our framework, which may further improve the performance of in-context learning on the FewMNER task. This is a direction for future efforts.

- Unfortunately, due to API limitations, we are unable to obtain results from the more powerful GPT4 model, which has a multimodal function. Further experiments and analysis are required.

We believe that addressing the above limitations can further improve the performance of our framework.

## Acknowledgements

This work was partially supported by the National Natural Science Foundation of China 62006062 and 62176076), Natural Science Foundation of GuangDong 2023A1515012922, Shenzhen Foundational Research Funding JCYJ20200109113441941, Guangdong Provincial Key Laboratory of Novel Security Intelligence Technologies 2022B1212010005.

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

# A  Appendix

## A.1  Impact of Different Output Format

| Methods | Twitter-2015 | | |
|---|---|---|---|
| | P(%) | R(%) | F1(%) |
| sentence | 51.10 | 51.84 | 51.47 |
| dictionary | 48.87 | 65.87 | 56.11 |

Table 10: Impact of different output format in 4-shot setting on $\mathcal{D}_{50}$ set of Twitter-2015.

| Methods | Twitter-2017 | | |
|---|---|---|---|
| | P(%) | R(%) | F1(%) |
| sentence | 61.41 | 61.73 | 61.57 |
| dictionary | 65.58 | 74.46 | 69.74 |

Table 11: Impact of different output format in 4-shot setting on $\mathcal{D}_{50}$ set of Twitter-2017.

To explore the influence of different output formats, we perform experiments comparing dictionary-based output format and sentence-based output format and show results in Table 10-11. We can obverse that compared with the sentence-based output format, the dictionary-based output format achieves 4.64% and 8.17% higher F1 in 4-shot setting on $\mathcal{D}_{50}$ set of Twitter2015 and Twitter2017 datasets, respectively, which demonstrates the dictionary-based output format is more suitable for in-context learning paradigm on ChatGPT. Perhaps this is related to the pre-training of ChatGPT on code, as this code-like output format is more comprehensible for ChatGPT.

| Replace Proportion | Twitter2015 | | | Twitter2017 | | |
|---|---|---|---|---|---|---|
| | P(%) | R(%) | F1(%) | P(%) | R(%) | F1(%) |
| 0% | **56.43** | **61.10** | **58.67** | **72.51** | **71.06** | **71.78** |
| 10% | 56.32 | 60.85 | 58.50 | 70.87 | 70.76 | 70.81 |
| 50% | 52.88 | 58.97 | 55.76 | 67.05 | 68.39 | 67.72 |
| 100% | 51.39 | 58.54 | 54.73 | 61.65 | 61.88 | 61.77 |

Table 12: Different labels replace proportion in 4-shot setting on $\mathcal{D}_{50}$ set. Note that the results presented here differ from the main experiment of paper due to OpenAI updating ChatGPT.

## A.2 Perturbation Analysis

We conduct a perturbation analysis to examine the effect of label noise on in-context learning. We randomly replace different proportions of labels with other labels in the examples and measure the impact of label noise on the F1 score of our framework. The results are shown in Table 12. As expected, the F1 score decreases as the label replacement ratio increases. When all the example labels are replaced (*i.e.*, 100%), the F1 score drops by 3.94% and 10.01% on two datasets, respectively. However, the F1 score does not drop to zero even when all the example labels are replaced, which indicates that the LLM has a strong intrinsic ability for this task and is not completely lost by the examples.

## A.3 Analyze Our Framework from Task Perspective

We here provide an analysis of why our framework performs well from the perspective of the FewMNER task. First, we model the FewMNER task as a generative task, which is more direct than the baseline methods that model the MNER task as a sequence labeling task. Sequence labeling methods not only require fine-grained prediction but also need to learn the association between different labels. Second, our framework utilizes image caption as a bridge to alleviate the gap between image modality and text modality, which can skip the process of information interaction between different modalities. Third, the four category concepts of people, location, organization, and miscellaneous in FewMNER tasks can be understood by LLM. Our framework can be readily adapted to the task objectives. For these reasons, our framework achieves good performance on the FewMNER task under the condition of small samples.