# OpenReview forum: "In-context Learning for Few-shot Multimodal Named Entity Recognition"
_EMNLP/2023/Conference — EMNLP 2023 Findings_

### Official Review · Reviewer_HAWy · 2023-07-30

**Soundness:** 2

**Excitement:**

2: Mediocre: This paper makes marginal contributions (vs non-contemporaneous work), so I would rather not see it in the conference.

**Paper Topic And Main Contributions:**

This paper focuses on the multimodal named entity recognition (MNER) task, which aims to detect named entities in a given text with the assistance of additional images. Furthermore, it concentrates on the field of few-shot multimodal named entity recognition (FewMNER) task, which is set in a scenario where only a few labeled text-image pairs are available for training.
The contributions of this paper are as follows:
1. This paper is the first paper to propose the FewMNER task and explore the potential of the in-context learning paradigm for this task.
2. To adapt the in-context learning paradigm to the FewMNER task, the paper addresses three related problems including the LLMs do not receive visual inputs, selecting useful examples for ICL in multimodal scenarios has not been approached and the absence of good demonstration designing. And they proposed a framework to accomplish this task, which includes:
- a retrieve module, which calculates the sum of text cosine similarity and image cosine similarity between test examples and candidate examples. The paper selects the top-k examples of the similarity ranking (k-nearest) as the examples to be used in ICL.
- a demonstration designing module including instruction construction, which helps the LLM understand the task and the entity labels, and demonstration construction, which contains image description, sentence , and output (ICL examples).
- a predict module which is simply feed the concatenation of instruction, demonstration and the test sample into LLM and get predicted output sequence.
3. Through comparison with previous competitive methods, the framework in this paper exhibits a significant advantage in this task. The authors conduct extensive analysis experiments to reveal the impact of various factors on its performance and provide novel insights for future research.

**Questions For The Authors:**

1. What are the practical applications of MNER, or in what scenarios can it be applied?
2. Have you tried any other task demonstrations (Instruction construction in your paper) and executed a more rigorous comparison among these prompts?


**Reasons To Accept:**

This paper is well-written and very easy to follow, with few grammar mistakes or typos.

**Reasons To Reject:**

This paper appears to be poorly lacking in innovation. It seems to me that in the era of LLM, anyone who wants to utilize LLMs to accomplish MNER task is very likely be able to carry out the research described in this paper in a method quite similar to the method outlined here. The reason is as follows:
1. Speaking of few-shot settings for any task including MNER, people naturally come up with the scenarios where only a small number of training samples are available. So I cannot really understand why the author claims that they first propose the FewMNER task. I mean it could be a consensus among all researchers in AI field, or one could say that the author is simply the first one to put this concept into a paper, but the concept itself is known by anybody. And anyone will convert the images to natural languages if he/she wants to use LLMs in this task, so the point of utilizing a image caption model of others cannot be considered as an innovation either.
2. Another thing is that the framework for FewMNER is too simple. You can find a large number of researches or implementations which use LLM and ICL to accomplish NLP tasks, and the general system architecture is almost the same with the one illustrated in this paper. The solution is simply to prepare a task-specified prompt and choose the best ICL examples  (in many cases, the most similar ones) as context.
3. I had expected a brand new method of selecting ICL examples in multimodal scenarios, but the one provided is kind of disappointing, as it simply calculates the sum of cosine similarities between images and texts.

**Reproducibility:**

5: Could easily reproduce the results.

**Reviewer Confidence:**

4: Quite sure. I tried to check the important points carefully. It's unlikely, though conceivable, that I missed something that should affect my ratings.

---

> ### Author Rebuttal · Authors · 2023-08-28
>
> We truly appreciate the time and effort you've taken to review our work.
>
> We thank the reviewer for the valuable comments. Before we detail the response, we would like to clear our motivation and contribution. As described in section Introduction, we are the first to investigate MNER from a few-shot perspective and validate the feasibility of ICL for the FewMNER task. Besides, we propose a complete framework to solve the FewMNER task, analyze the impact of ICL on performance, and achieve satisfactory results. We sincerely hope that these clarifications provide a clearer understanding of the essence of our work. We kindly request that if you have any specific questions or suggestions pertaining to the contributions of our paper, please share them with us. Your expertise and insights are invaluable to our work.
>
> **Comment 1**
>
> >Speaking of few-shot settings for any task including MNER, people naturally come up with the scenarios where only a small number of training samples are available. So I cannot really understand why the author claims that they first propose the FewMNER task. I mean it could be a consensus among all researchers in AI field, or one could say that the author is simply the first one to put this concept into a paper, but the concept itself is known by anybody. And anyone will convert the images to natural languages if he/she wants to use LLMs in this task, so the point of utilizing a image caption model of others cannot be considered as an innovation either.
>
> **Response:**
>
> We apologize for the confusion caused by our expression. To answer your confusion, here we explain why we claim that we first propose the FewMNER task. The reason is that the definition of our proposed FewMNER is different from previous few-shot tasks, which usually construct N-way K-shot setting to utilize meta-learning methods to solve few-shot tasks. Our proposed FewMNER task utilizes only K-shot. We want to distinguish it from previous work and emphasize our new setting. After our serious discussion, we think you are right and agree with you. Thank you for your thoughtful and valuable feedback. We decide to revise our expression in the next version.
>
>
> In this paper, our main contribution is to validate the feasibility of ICL for the FewMNER task, propose a complete framework to solve the FewMNER task and achieve satisfactory results. The image caption is a sub-module of our framework, which we use to overcome the limitation of LLM that only accepts natural language as input.
>
> ------
>
> **Comment 2**
>
> >Another thing is that the framework for FewMNER is too simple. You can find a large number of research or implementations which use LLM and ICL to accomplish NLP tasks, and the general system architecture is almost the same with the one illustrated in this paper. The solution is simply to prepare a task-specified prompt and choose the best ICL examples (in many cases, the most similar ones) as context.
>
> **Response:**
>
> Thank you for your feedback. We agree with your viewpoint that there are many studies on the in-context learning (ICL) of LLM. However, most of them solve generation and classification problems, and solving extraction tasks in multimodal scenarios is still a challenge. Regarding our framework, we focus on multimodal and extraction characteristics to design effective modules (e.g., similarity rank, and output format).
>
> - For mitigating the bias of different modality models, we propose an efficient selection method based on text and image similarity ranks. Comparing sorting based on the sum of image and text similarity ranks and sorting based on the sum of image and text similarity scores, the former outperforms the F1 by 2.03% (54.08%->56.11%) and 2.67% (67.07%->69.74%) in 4-shot setting on $\mathbf{\mathcal{D}_{50}}$ set of two datasets, respectively (see w/ score in Table 3).
>
> - For adapting the extraction task, we design a dictionary-based format that explicitly defines the output structure as `{'PER':[], "ORG":[], "LOC":[], "MISC":[]}`. Compared with the sentence-based output format, the dictionary-based output format achieves 4.64% (51.47%->56.11%) and 8.17% (61.57%->69.74%) higher F1 in 4-shot setting on $\mathbf{\mathcal{D}_{50}}$ set of two datasets, respectively (see Table 10-11 in paper).
>
> We hope that these effective and purposefully designed modules can differentiate our work from previous work. Additionally, we believe the simple and effective framework is helpful for future work.
>
> --------
>
> **Comment 3**
>
> >I had expected a brand new method of selecting ICL examples in multimodal scenarios, but the one provided is kind of disappointing, as it simply calculates the sum of cosine similarities between images and texts.
>
> **Response:**
>
> Thank you for your constructive feedback. To demonstrate the effectiveness of our method, we compare it with a new method for selecting ICL examples. This method is from the paper: [Automatic Chain of Thought Prompting in Large Language Models (ICLR2023)](https://openreview.net/forum?id=5NTt8GFjUHkr). Following this paper, we first cluster text or image features into K clusters. Then, we sample the cluster center of each cluster, which aims to enhance the diversity of ICL examples. The experimental results are as follows. From the result, we can see that this clustering method performs lower than our method. This result demonstrates that our method is simple and effective.
>
> | Methods               | Twitter2015(F1%) | Twitter2017(F1%) |
> | --------------------- | :--------------: | :--------------: |
> | **Ours**              |      58.67       |      71.78       |
> | Cluster (Text)        |      57.83       |      71.35       |
> | Cluster (Image)       |      57.38       |      71.09       |
> | Cluster (Text, Image) |      57.19       |      71.10       |
>
> *Table 1: Different select ICL example methods in 4-shot setting on $\mathbf{\mathcal{D}_{50}}$ set. Note that the results presented here differ from paper due to OpenAI updating ChatGPT.*
>
> -----
>
> **[Q1]** What are the practical applications of MNER, or in what scenarios can it be applied?
>
> **[A1]** The MNER task can serve as the foundation for more advanced tasks, such as multimodal event extraction [1], multimodal aspect-based sentiment analysis [2], and building multimodal knowledge graphs [3]. For example, by using MNER, we can identify the entities that are related to a specific event or topic from social media data, and then analyze the sentiment, opinion, or attitude of the users towards those entities.
>
> [1]: [Joint Multimedia Event Extraction from Video and Article](https://aclanthology.org/2021.findings-emnlp.8) (Chen et al., EMNLP 2021)
>
> [2]: [AoM: Detecting Aspect-oriented Information for Multimodal Aspect-Based Sentiment Analysis](https://aclanthology.org/2023.findings-acl.519) (Zhou et al.,ACL 2023)
>
> [3]: [MMEKG: Multi-modal Event Knowledge Graph towards Universal Representation across Modalities](https://aclanthology.org/2022.acl-demo.23) (Ma et al., ACL 2022)
>
> **[Q2]** Have you tried any other task demonstrations (Instruction construction in your paper) and executed a more rigorous comparison among these prompts?
>
> **[A2]** Yes, when designing the framework, we tested the performance of different instructions and chose the one that performed better. To answer your concerns, we design four types of instruction and compare their performance. The details are as follows. From the experimental results, we can see that more detailed instruction descriptions lead to better performance.
>
> - Instruction (paper): You are a smart and intelligent Multimodal Named Entity Recognition (MNER) system. I will provide you the definition of the entities you need to extract, the sentence from where you extract the entities, the image description from image associated with sentence and the output format with examples.
>
> - Instruction (v2): Please extract the entities from text.
>
> - Instruction (v3): Do multimodal named entity recognition task.
>
> | Instruction     | Twitter2015(F1%) | Twitter2017(F1%) |
> | --------------- | :--------------: | :--------------: |
> | Instruction (paper)         |      58.67       |      71.78       |
> | Instruction (v2) |      57.34       |      71.91       |
> | Instruction (v3) |      57.72       |      70.27       |
> | No Instruction  |      55.60       |      71.03       |
>
> *Table 2: Different instruction in 4-shot setting on $\mathbf{\mathcal{D}_{50}}$ set. Note that the results presented here differ from paper due to OpenAI updating ChatGPT.*

---

### Official Review · Reviewer_BB5Z · 2023-08-03

**Soundness:** 4

**Excitement:**

3: Ambivalent: It has merits (e.g., it reports state-of-the-art results, the idea is nice), but there are key weaknesses (e.g., it describes incremental work), and it can significantly benefit from another round of revision. However, I won't object to accepting it if my co-reviewers champion it.

**Paper Topic And Main Contributions:**

This paper introduces the FewMNER task for the first time and investigates the potential of the in-context learning paradigm in addressing this task. To adapt the in-context learning paradigm to the FewMNER task, this paper addresses three related problems and proposes a framework to accomplish this task. Through comparison with previous competitive methods, this paper's framework exhibits a significant advantage in this task. This paper also conducts extensive analysis experiments to reveal the impact of various factors on its performance and provides novel insights for future research.

**Reasons To Accept:**

1. This is the first attempt to combine few-shot learning with multimodal named entity recognition (NER), introducing the FewMNER task, and this work is also the first exploration of the potential of in-context learning in FewMNER.
2. This work conducts comprehensive experiments to validate the effectiveness of the proposed algorithm.
3. This paper is well-written and well-organized, and is easy to follow.

**Reasons To Reject:**

1. This work appears to lean more towards an applied paper rather than an academic research paper. It extensively relies on existing techniques (e.g., Image caption model and ChatGPT), with limited original contributions.
2. While the experimental results are promising, the credit for the success largely goes to the Large Language Model (LLM) employed. The paper, however, lacks an in-depth analysis from the perspective of MNER to explain why the performance is good, resulting in limited interpretability.
3. Some recent relevant works have not been compared, such as:  1). M3S: Scene Graph Driven Multi-Granularity Multi-Task Learning for Multi-Modal NER;  2). Reducing the Bias of Visual Objects in Multimodal Named Entity Recognition.


**Reproducibility:**

3: Could reproduce the results with some difficulty. The settings of parameters are underspecified or subjectively determined; the training/evaluation data are not widely available.

**Reviewer Confidence:**

3: Pretty sure, but there's a chance I missed something. Although I have a good feel for this area in general, I did not carefully check the paper's details, e.g., the math, experimental design, or novelty.

---

> ### Author Rebuttal · Authors · 2023-08-28
>
> We truly appreciate the time and effort you've taken to review our work.
>
> **Comment 1**
>
> >This work appears to lean more towards an applied paper rather than an academic research paper. It extensively relies on existing techniques (e.g., Image caption model and ChatGPT), with limited original contributions.
>
> **Response**:
>
> Thanks for your comment. Although our framework leverages existing techniques (e.g., Image caption model and ChatGPT), we maintain that our work has academic value. The specific reasons are as follows:
>
> - Many studies have explored the in-context learning (ICL) of LLM, but they mostly focus on generation and classification problems. Solving extraction tasks in multimodal scenarios is still a challenge. In this paper, our main contribution is to validate the feasibility of ICL for FewMNER task and analyze the impact of ICL on the performance of FewMNER task.
> - For the framework, we design simple and effective modules (e.g., retrieve-based method, similarity rank, output format) according to multimodal and extraction task characteristics, which have a good generalization and can be easily applied to other multimodal extraction tasks (e.g., multimodal event extraction).
> - Besides the framework, we focus on the demonstration designing module and conduct extensive experiments to obtain interesting findings, e.g., selecting similar examples, sorting them in ascending order, and using more examples can improve the performance of ICL on FewMNER task. Furthermore, we perform error analysis and observe that increasing the number of examples reduces boundary errors but not category errors. These findings can provide novel insights and guidance for feature work on FewMNER and ICL.
>
> We hope these expositions can demonstrate the academic value of our work.
>
> -------
>
> **Comment 2**
>
> >While the experimental results are promising, the credit for the success largely goes to the Large Language Model (LLM) employed. The paper, however, lacks an in-depth analysis from the perspective of MNER to explain why the performance is good, resulting in limited interpretability.
>
> **Response**:
>
> Sorry for the lack of this analysis. To answer your concerns, we here provide an analysis on why our framework performs well from the perspective of FewMNER task. First, we model the FewMNER task as a generative task, which is more direct than the baseline methods that model the MNER task as a sequence labeling task. Sequence labeling methods not only require fine-grained prediction but also need to learn the association between different labels. Second, we utilize image caption as a bridge to alleviate the gap between image modality and text modality, which can skip the process of information interaction between different modalities. Third, the four category concepts of people, location, organization, and miscellaneous in FewMNER tasks can be understood by LLM. Our framework can be readily adapted to the task objectives. For these reasons, our framework achieves good performance on the FewMNER task under the condition of small samples. We appreciate your valuable suggestion and we will include this analysis in the next version of the paper.
>
> -------
>
> **Comment 3**
>
> >Some recent relevant works have not been compared, such as: 1). M3S: Scene Graph Driven Multi-Granularity Multi-Task Learning for Multi-Modal NER; 2). Reducing the Bias of Visual Objects in Multimodal Named Entity Recognition.
>
> **Response:**
>
> Thank you for your thoughtful reminder of these two articles. Unfortunately, we could not find the open source for the paper 1). M3S: Scene Graph Driven Multi-Granularity Multi-Task Learning for Multi-Modal NER. We are trying to reproduce this paper ourselves, but we have not completed it yet due to time constraints. Luckily, we obtain the code for the paper 2). Reducing the Bias of Visual Objects in Multimodal Named Entity Recognition, namely DebiasCL. The experimental results are as follows. We can observe that our framework still significantly outperforms the DebiasCL methods on $\mathbf{\mathcal{D} _{10}}$, $\mathbf{\mathcal{D} _{50}}$, and $\mathbf{\mathcal{D} _{100}}$ sets.
>
>
> | Setting | Method   | $\mathbf{\mathcal{D}_{10}}$(F1%) | $\mathbf{\mathcal{D}_{50}}$(F1%) | $\mathbf{\mathcal{D}_{100}}$(F1%) | $\mathbf{\mathcal{D}_{all}}$(F1%) |
> | ------- | -------- | :------------------------------: | :------------------------------: | :-------------------------------: | :-------------------------------: |
> |       | UMT      |               0.79               |              40.09               |               52.66               |               73.41               |
> |         | UMGF     |               1.34               |              43.33               |               51.86               |               74.85               |
> |         | HVPNeT   |              20.17               |              48.46               |               57.94               |               75.32               |
> | FT        | DebiasCL |               1.74               |              39.91               |               50.44               |               75.28               |
> | 8-shot  | **Ours** |              56.57               |              56.15               |               55.45               |               58.14               |
>
> *Table 1: The experiment results on Twitter2015.*
>
> | Setting | Method   | $\mathbf{\mathcal{D}_{10}}$(F1%) | $\mathbf{\mathcal{D}_{50}}$(F1%) | $\mathbf{\mathcal{D}_{100}}$(F1%) | $\mathbf{\mathcal{D}_{all}}$(F1%) |
> | ------- | -------- | :------------------------------: | :------------------------------: | :-------------------------------: | :-------------------------------: |
> |       | UMT      |               3.81               |              57.14               |               64.11               |               85.31               |
> |         | UMGF     |               1.65               |              56.50               |               65.01               |               85.51               |
> |         | HVPNeT   |              33.58               |              58.63               |               65.14               |               86.87               |
> |  FT       | DebiasCL |               3.36               |              41.45               |               67.83               |               86.84               |
> | 8-shot  | **Ours** |              69.11               |              70.00               |               72.97               |               73.26               |
>
> *Table 2: The experiment results on Twitter2017.*
>
> We will continue to reproduce the code of the first paper (i.e., M3S) and supplement this comparison result in the next version.

---

### Official Review · Reviewer_DFxJ · 2023-08-05

**Soundness:** 3

**Excitement:**

4: Strong: This paper deepens the understanding of some phenomenon or lowers the barriers to an existing research direction.

**Paper Topic And Main Contributions:**

The main focus of the paper is on the few-shot Multimodal Named Entity Recognition (FewMNER) task, specifically, locating and identifying named entities for a text-image pair only using a small number of labeled examples. This paper proposes an in-context learning paradigm to leverage a large language model to solve FewMNER. Especially, for three potential problems which may hinder the ICL paradigm, this paper first leverages the image caption model to generate textual description; second, they construct a similarity rank for multimodal; and they add the description of entity category meaning to the instruction and feed the concatenation of instruction, demonstration and test sample to LLM for predicting the entity tags. The main contribution of the article is to provide a solution paradigm for MNER in a few-shotscenario.
Around this In-context learning paradigm, the authors' study provides certain insights.

**Questions For The Authors:**

A. Are there other metrics for calculating the Cos distance of image representations?

B. What is the few-shot setting in sampling MNER instances? For example, when a sample contains multiple entities, it is unclear whether it should be considered as one sample or multiple samples in the sampling process. If it is counted as one, it is also unclear how to ensure that each entity category occurs only once in the k-shot setting.

C. We know that the construction of Demonstration is very important for ICL, have the authors considered whether slight perturbations can enhance ICL or affect it, and these aspects of consideration and analysis would be interesting to see.

D. How can we avoid that LLM has learned all the data in the dataset during the pretraining process, how do the authors see or avoid this problem?


**Reasons To Accept:**

- Currently, the FewMNER problem receives not much attention, however, it remains significant in the domains of multimodal and NER processing.

- The paper presents a potentially promising approach for addressing the FewMNER task through In Context Learning and LLM usage from a limited number of examples without the need for training.

- The comparative experiments are adequate, and the results also show that the method proposed in the paper is competitive.

**Reasons To Reject:**

- The data comparison is solely based on a single source, specifically social media datasets from Twitter. However, the results and the model's ability to generalize are not strongly supported.

- The proposed method is primarily based on LLM and ICL. However, it heavily depends on the accuracy of image-caption in the detailed processing, which can lead to the problem of error propagation.

- Although the primary focus of the paper is on ICL, the essential procedures involved, such as retrieve-based methods and similarity rank, are general in V+L models. However, the ablation experiments fail to distinguish the contribution of these aspects, which undermines the overall credibility of the main innovation presented in the paper.
- The main focus in this paper is not clearly addressed when it comes to few-shot learning. There have been previous studies on few-shot named entity recognition (FNER), but this paper is confusing because it does not provide clear guidance on how to sample in few-shot MNER.



**Reproducibility:**

5: Could easily reproduce the results.

**Reviewer Confidence:**

4: Quite sure. I tried to check the important points carefully. It's unlikely, though conceivable, that I missed something that should affect my ratings.

---

> ### Author Rebuttal · Authors · 2023-08-28
>
> We truly appreciate the time and effort you've taken to review our work.
>
> **Comment 1**
>
> >The data comparison is solely based on a single source, specifically social media datasets from Twitter. However, the results and the model's ability to generalize are not strongly supported.
>
> **Response**:
>
> Thanks for this insightful comment. Indeed, using datasets from other sources to support the performance and generalizability of the proposed framework is required. However, to the best of our knowledge, there are currently only two publicly available datasets (i.e., Twitter2015 and Twitter2017) for the MNER task. This is why we only utilize these two datasets to evaluate the proposed framework in this paper. Besides, we are annotating a new MNER dataset, which covers several different social platforms. We will conduct research on this diverse dataset in the future.
>
> --------
>
> **Comment 2**
>
> >The proposed method is primarily based on LLM and ICL. However, it heavily depends on the accuracy of image-caption in the detailed processing, which can lead to the problem of error propagation
>
> **Response**:
>
> Thanks a lot for this interesting comment. We agree with your viewpoint that if the generated caption for each image is not quite accurate, the image-caption module would cause errors to propagate through our framework. In preliminary experiments, we randomly selected 100 samples to evaluate the quality of image captions. We found that most of captions are accurate and factual, so this error is small.
>
> ------
>
> **Comment 3**
>
> >Although the primary focus of the paper is on ICL, the essential procedures involved, such as retrieve-based methods and similarity rank, are general in V+L models. However, the ablation experiments fail to distinguish the contribution of these aspects, which undermines the overall credibility of the main innovation presented in the paper.
>
> **Response**:
>
> We apologize for not making it clear. To distinguish the contribution of the retrieve-based methods proposed in this work, we compare the similar, dissimilar, and random retrieve methods and conduct experiments in sub-section Different Examples Analysis (see Table 5). The results show that the similar method achieves the best performance, followed by the random method, and the dissimilar method performs the worst.
>
> For similarity rank, we present results in the ablation study (see w/ score in Table 3). Here, w/ score indicates that examples are selected based on the sum of image and text similarity scores. From this table, we can see that sorting based on the sum of image and text similarity ranks outperforms that of image and text similarity scores (i.e., w/ score).
>
> We hope that these experimental comparisons can distinguish the contribution of these aspects. To make it more convincing and clearer, we will amend the relevant text description.
>
> -------
>
> **Comment 4**
>
> >The main focus in this paper is not clearly addressed when it comes to few-shot learning. There have been previous studies on few-shot named entity recognition (FNER), but this paper is confusing because it does not provide clear guidance on how to sample in few-shot MNER.
>
> **Response**:
>
> Very sorry for the unclear expression. To answer your confusion, we here provide clearer guidance on how to sample in a few-shot MNER. Different from the previous few-shot NER, which constructs an N-way K-shot setting, our proposed FewMNER task utilizes only K-shot, i.e., selects K samples from the training dataset and does not consider whether each entity category occurs N times.  To illustrate this sample, we provide an example in the Introduction. As shown in Figure 1(b), the 2-shot MNER task aims to accomplish MNER based on two labeled text-image pair examples. We hope that these explanations will give you a clearer understanding of how to sample in few-shot MNER.
>
> ---------
>
> **[Q1]** Are there other metrics for calculating the Cos distance of image representations?
>
> **[A1]** Yes, there are other metrics such as L1, L2 and Dot Product. We compare their performance on the FewMNER task. The experimental results are as follows:
>
> | Distance  Metrics | Twitter2015(F1%) | Twitter2017(F1%) |
> | ----------------- | :--------------: | :--------------: |
> | L1                |      59.16       |      71.90       |
> | L2                |      58.79       |      71.94       |
> | Dot Product       |      58.59       |      70.77       |
> | Cosine            |      58.67       |      71.78       |
>
> *Table 1: Different distance metrics in 4-shot setting on $\mathbf{\mathcal{D}_{50}}$ set. Note that the results presented here differ from paper due to OpenAI updating ChatGPT.*
>
> From the results, we can notice that the results of different metrics are similar, indicating that the effect of distance metrics is small. It is fine to select any reasonable distance metric. In our work, we originally chose cosine similarity because it is commonly used.
>
> **[Q2]** What is the few-shot setting in sampling MNER instances?
>
> **[A2]** This question is similar to **Comment 4**. We view one sample as 1-shot. For example, 2-shot utilizes only two labeled text-image pair examples. Different from the previous few-shot NER task that constructs N-way K-shot setting, our proposed FewMNER task utilizes only K-shot, i.e., selects K samples from the train dataset, and does not consider whether each entity category occurs N times. Due to the ICL paradigm, we do not need to construct N-way K-shot setting to ensure that each entity category appears only once. It is worth noting that few-shot baselines in the main experiment keep N-way K-shot setting since they apply meta-learning for training.
>
> **[Q3]** We know that the construction of Demonstration is very important for ICL, have the authors considered whether slight perturbations can enhance ICL or affect it, and these aspects of consideration and analysis would be interesting to see.
>
> **[A3]** Thanks a lot for your very interesting question. In this paper, we conduct experiments to explore the impact of example sort (see Table 6). We use the same examples and apply different sorting methods, i.e., ascending, descending, and random. The results show that the ascending sort method slightly outperforms the other methods. This suggests that different orderings of the same examples have little effect on ICL.
>
> Thanks for the reminder, we perform another perturbation experiment. We randomly replace different proportions of labels with other labels to explore the impact of misinformation on ICL. The experimental results are as follows:
>
> | Replace Proportion | Twitter2015(F1%) | Twitter2017(F1%) |
> | ------------------ | :--------------: | :--------------: |
> | 0%                 |      58.67       |      71.78       |
> | 10%                |      58.50       |      70.81       |
> | 50%                |      55.76       |      67.72       |
> | 100%               |      54.73       |      61.77       |
>
> *Table 2: Different labels replace proportion in 4-shot setting on $\mathbf{\mathcal{D}_{50}}$ set. Note that the results presented here differ from paper due to OpenAI updating ChatGPT.*
>
> As the label replacement ratio increases, F1 will continue to decline. Comparing replacing all example labels (100%) with not replacing examples (0%), the former reduces the F1 by 3.94% (58.67%->54.73%) and 10.01% (71.78%->61.77%) on two datasets, respectively. Furthermore, we can notice that the F1 score does not drop to 0 when replacing all example labels (100%), which shows that the LLM has strong self-ability for this task and is not completely lost by the examples. This perturbation experiment is very interesting, thanks again for your suggestion. We will supplement this part of the experiment in the next edition of the paper.
>
> **[Q4]** How can we avoid that LLM has learned all the data in the dataset during the pretraining process, how do the authors see or avoid this problem?
>
> **[A4]** Thank you for your insightful question and your interest in our work. This is a very interesting question that got me thinking. Here, We share our viewpoint. We can simply split the ability of LLM into two aspects: self-ability and incremental ability. Self-ability can be viewed as the performance of LLM on zero-shot learning, which does not use any examples from the target domain. Incremental ability can be viewed as an improvement of LLM on K-shot learning, which uses a few examples from the target domain as demonstration.
>
> We hypothesize that if LLM has learned all the data in the dataset during the pretraining process, then adding more examples from the same dataset would not have much impact on its performance. However, as shown in Table 9, we observe that the F1 score generally increases with the number of examples on both Twitter2015 and Twitter2017 datasets. For example, comparing 0-shot and 32-shot, the F1 scores are 41.61% vs. 55.92% and 57.92% vs. 71.76% on the two datasets, respectively. According to the above analysis, I tend that LLM has not learned all the data in the dataset during the pretraining process and can still benefit from incremental learning with demonstration.

---

### Meta-Review · Area_Chair_i8wL · 2023-09-24

**Recommendation:** 3

**Metareview:**

The paper builds a few-shot learning based model for multimodal NER.  There has been a very little attempts in this area, and hence it makes a contributions. There are certain aspects that need to be addressed: few shot setting for the NER; use of image captions and its impact on the overall model performance, and the use of ChatGPT in proper way.

---

### Decision · Program_Chairs · 2023-10-07

**Decision:**

Accept-Findings

**Comment:**

The paper builds a few-shot learning based model for multimodal NER.  There has been a very little attempts in this area, and hence it makes a contributions. There are certain aspects that need to be addressed: few shot setting for the NER; use of image captions and its impact on the overall model performance, and the use of ChatGPT in proper way.